# Evaluating Structural Transformation and Conceptual Abstraction in DeepSeek V3 on ConceptARC

## Abstract

This paper evaluates the capabilities of DeepSeek V3 on a structured abstraction benchmark we call *ConceptARC*, an organisation of Abstraction and Reasoning Corpus (ARC) tasks into sixteen explicit conceptual families. Our experimental protocol measures two complementary capacities: (1) procedural transformation performance (exact output accuracy) and (2) conceptual identification (concept classification accuracy). We describe an automated, reproducible evaluation pipeline, report aggregated results across concept families, analyse representative failure modes, and discuss theoretical implications for compositional generalisation. The main empirical finding is a consistent dissociation: DeepSeek V3 often attains moderate procedural competence yet exhibits unstable and comparatively weak concept-level abstraction, motivating hybrid approaches that couple object-centric symbolic structure with neural perception.

## 1 Introduction

The Abstraction and Reasoning Corpus (ARC) (Chollet, 2019) and ConceptARC (Mitchell, 2023) are designed to probe systematic generalisation and abstract reasoning through a diverse set of small-grid transformation tasks (Figure 1). To sharpen analysis of concept-level generalisation, we organise ARC tasks into sixteen conceptual families (*AboveBelow*, *Center*, *CleanUp*, *CompleteShape*, *Copy*, *Count*, *ExtendToBoundary*, *ExtractObjects*, *FilledNotFilled*, *HorizontalVertical*, *InsideOutside*, *MoveToBoundary*, *Order*, *SameDifferent*, *Scale*, and *TopBottom* variants). Each family corresponds to a transformation principle intended to test structural abstraction rather than superficial pattern matching.

This study asks two related questions: (i) can DeepSeek V3 infer procedural rules from minimal exemplars and apply them to novel inputs? and (ii) can it situate inferred rules within a stable conceptual taxonomy? We evaluate these questions by separately measuring correctness of predicted outputs and correctness of concept classification, enabling a diagnostic separation between procedural competence and conceptual abstraction.

## 2 Experimental framework and evaluation methodology

### 2.1 Dataset and concept families

ConceptARC groups ARC tasks into sixteen concept families described above. For each family we selected ten tasks (randomly sampled with stratification to preserve within-family variance), yielding a balanced set for cross-family comparison.

### 2.2 Evaluation pipeline and prompting protocol

For every selected task the pipeline provided DeepSeek V3 with: all available training input–output pairs and a single unseen test input grid. The model was instructed to (a) infer the transformation rule, (b) produce a predicted output grid for the test input, (c) select exactly one concept label from

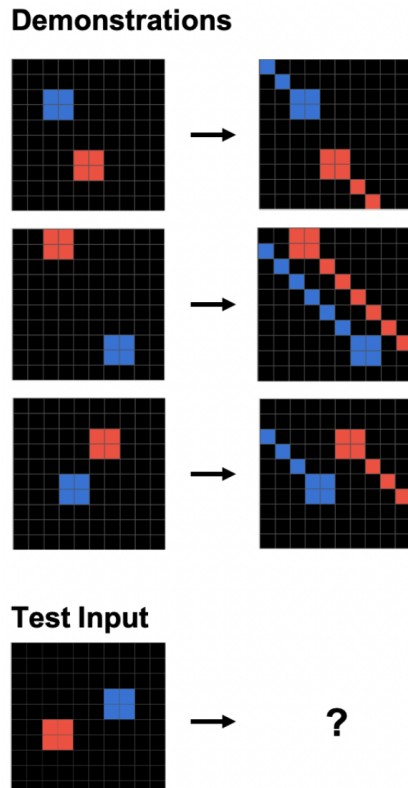

Figure 1: Example of a ConceptARC task. The task in ? needs to be solved.

the predefined list of sixteen families, and (d) return a confidence estimate. Responses were restricted to a structured JSON format to permit deterministic parsing and automated scoring.

## 2.3 METRICS

Two primary metrics were computed per task:

1. **Exact output accuracy:** strict element-wise equality between the predicted and ground-truth output grids (binary per task).

2. **Concept classification accuracy:** whether the model's selected concept label matched the dataset taxonomy's ground-truth family.

We treat these metrics independently. Exact output accuracy captures procedural competence; concept classification accuracy captures stability of conceptual abstraction across structurally related tasks.

## 3 RESULTS

Across the sixteen concept families DEEPSEEK V3 showed moderate procedural performance but substantially weaker concept classification. Aggregating tasks by family, exact output accuracy frequently approached roughly 50% in families where transformations were visually salient and geometric (e.g., explicit symmetry, uniform separators, or straightforward propagation). In contrast, classification accuracy was lower and exhibited greater volatility across runs.

### 3.1 REPRESENTATIVE FAMILY-LEVEL OBSERVATIONS

Below we summarise recurring patterns observed during evaluation.

**AboveBelow:** Reliable when horizontal separators are uniform and unambiguous. Performance suffered when the separator participated in conditional modifications; concept confusion with *ExtendToBoundary* and *MoveToBoundary* was common.

**Center:** Accurate when centrality was visually explicit (symmetry or perceptual midpoint). Tasks requiring computed central placement saw degraded performance.

**CleanUp:** Solved when noise removal reduced to majority-colour filtering. Connected-component or context-sensitive noise removal posed difficulty.

**CompleteShape:** Mirroring and overt symmetric completion were often correct; reconstruction requiring latent-axis inference or occlusion reasoning produced errors.

**Copy:** Straightforward duplication (horizontal/vertical) succeeded; conditional or selective copying led to overgeneralisation.

**Count:** Moderately good at counting contiguous blocks; failures when counting required distinguishing multiple connected components or relational filters.

**ExtendToBoundary / MoveToBoundary:** Performed well for obvious directional propagation. Classification instability persisted: tasks were sometimes mislabelled as *Copy* or other spatial families.

**ExtractObjects / FilledNotFilled:** Partial object segmentation capability; bounding-box precision and strict delimitation were inconsistent.

**InsideOutside:** Containment reasoning was fragile, especially with overlapping or hierarchical boundaries.

**Order / SameDifferent:** Relational abstractions produced high variability in both execution and label prediction.

**Scale:** One of the most challenging families when resizing required proportional inference rather than simple duplication.

**TopBottom variants:** Performed adequately when vertical symmetry was explicit; degraded when hidden balance or perspective inference was needed.

Overall, DEEPSEEK V3 is robust when transformations align with visually salient geometric operations, and fragile when tasks demand compositional reasoning, conditional constraints, connected-component analysis, or precise spatial delimitation.

## 4 ERROR ANALYSIS

We identify four dominant failure modes that recur across families:

1. **Concept confusion:** structurally similar transformations are conflated, producing unstable category boundaries.

2. **Overgeneralisation (scope errors):** inferred rules applied globally rather than respecting intended local scope.

3. **Boundary misalignment:** small spatial offsets or overextensions that prevent exact equality despite structural similarity.

4. **Weak object segmentation:** unreliable connected-component extraction and imprecise bounding delineation.

These modes together indicate a reliance on surface-level geometric heuristics rather than systematic, symbolic abstraction. The model's behaviour aligns with observations from the ARC literature: neural and prompted systems often succeed at interpolative pattern completion but struggle with systematic compositional generalisation.

## 5 THEORETICAL IMPLICATIONS

The dissociation between procedural execution and concept identification is theoretically consequential. Correct output generation does not imply formation of a stable, generalisable conceptual

representation. For robust abstraction one must encode invariant transformation principles that persist across variations in surface presentation. DEEPSEEK V3's instability suggests probabilistic pattern alignment dominates its internal strategy rather than discrete category-based representations.

From an architectural standpoint, these results motivate hybrid neuro-symbolic solutions: explicit object-centric representations, transformation libraries parameterised by symbolic constraints, and hierarchical abstraction mechanisms may improve stability. Object-centric pre-processing (segmentation to symbolic primitives) and downstream rule engines for scope and composition are promising directions.

## 6 DISCUSSION

Our evaluation provides a reproducible baseline for DEEPSEEK V3 on ConceptARC and a diagnostic account of its reasoning profile. Key takeaways are:

- Procedural competence is conditional: success depends on visual salience and the simplicity of the transformation.
- Concept-level abstraction is unstable: the model often fails to assign tasks to correct conceptual families despite producing near-correct outputs.
- Small spatial imprecisions frequently break exact-match scoring, exposing fragility in fine-grained structural reasoning.

Limitations of the present study include the finite sampling of tasks per family (ten tasks) and reliance on a single prompting protocol; future work should evaluate sensitivity to prompt design, evaluate larger and adversarially selected task sets, and measure within-family generalisation across broader morphological variation.

## 7 CONCLUSION

We presented a structured evaluation of DEEPSEEK V3 on ConceptARC, separating procedural execution from concept identification. The principal empirical finding is a persistent gap: the model attains moderate output accuracy under favourable visual conditions but fails to establish stable conceptual boundaries across families. This suggests that procedural correctness alone is an insufficient indicator of abstract reasoning. Achieving systematic generalisation on ARC-like problems likely requires integrating explicit structural representations and symbolic constraints with neural perception.

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
