# OpenReview forum: "EVALUATING STRUCTURAL TRANSFORMATION AND CONCEPTUAL ABSTRACTION IN DEEPSEEK V3 ON CONCEPTARC"
_mathai.club/MathAI/2026/Conference — MathAI 2026 Conference Submission_

### Official Review · Reviewer_zTNk · 2026-03-12
**Review of ``Evaluating Structural Transformation and Conceptual Abstraction in DeepSeek V3 on ConceptARC''**

**Rating:** 6
**Confidence:** 3

**Review:**

Summary:

This article evaluates the logical abilities of DeepSeek V3 on ConceptARC, a conceptually organized version of the Abstractions and Reasoning Corpus (ARC). The authors divide the ARC tasks into sixteen conceptual families and propose an evaluation protocol that separates the accuracy of the procedural transformation (the exact correspondence of the output data) from the conceptual abstraction (the accuracy of the classification of concepts). The main conclusion is the discrepancy between these two possibilities: the model often achieves moderate procedural accuracy with visually noticeable transformations, but demonstrates weaker and unstable abstraction at the conceptual level.

Strengths:

The article is clearly written and represents a reproducible evaluation algorithm. The distinction between procedural correctness and conceptual abstraction is a useful diagnostic perspective for ARC-style reasoning tasks. A qualitative analysis of the types of errors (for example, confusion in concepts, mismatch of boundaries and weak segmentation of objects) allows us to obtain useful information about the logical behavior of large language models.

Weaknesses:

The experimental assessment is limited in several aspects. First, the study evaluated only one model (DeepSeek V3), which makes it difficult to contextualize the results without comparing them with other models or ARC solvers. Secondly, the evaluation uses only ten tasks per family of concepts, which raises concerns about statistical reliability. Finally, the document uses a single query protocol and does not evaluate the sensitivity of the results to quick processing.

Questions

Can the authors make comparisons with other models within the same evaluation protocol?

How sensitive are the results to a specific sample of tasks (10 tasks per family)?

How often does the model produce correct outputs, but incorrect concept designations, and vice versa?

Are the results stable when using different hint strategies?

Conclusion:

Despite the fact that the article is clearly written and the assessment approach is interesting, empirical research is currently too limited to support it with convincing conclusions. In particular, the lack of basic comparisons and the small scale of evaluation make it difficult to fully assess the significance of the results obtained. With a more comprehensive assessment (including additional models and a broader range of tasks), this work could be a useful contribution, but in its current form it is somewhat below the acceptable level.

---

### Official Review · Reviewer_xrFv · 2026-03-13
**This article presents an experimental evaluation of the DeepSeek V3 model on the ConceptARC benchmark. The benchmark includes different sets of abstraction and analogy tasks. Each set assesses specific abstraction and generalization capabilities of the model.**

**Rating:** 6
**Confidence:** 4

**Review:**

However, the paper has several problems:
1) Sample size. Ten problems per family is too small to draw statistically significant conclusions.
2) Single prompt protocol. LLM performance is extremely sensitive to the prompt formulation. Without ablation studies on the prompt design, it is impossible to separate model limitations from prompting artifacts.
3) Lack of baselines. There is no comparison with other models, which makes it impossible to place the DeepSeek V3 results in the context of the current state of the field.
4) "Inter-run volatility" is mentioned but not quantified.

The four-page article appears unfinished for conference publication. There is no Related Work section, preventing the paper from being positioned relative to the existing literature. The bibliography contains only two references.

---

### Official Review · Reviewer_PDGT · 2026-03-13
**This paper presents a targeted evaluation of the DeepSeek V3 large language model on the ConceptARC benchmark, which is designed to test abstract reasoning and conceptual abstraction. The authors introduce a novel evaluation methodology that separates two distinct capabilities: procedural execution (can the model correctly transform an input grid to produce the correct output?) and concept identification (can the model correctly label which type of transformation it is performing?). The key finding is a significant dissociation between these two abilities: DeepSeek V3 achieves moderate success on procedural tasks under favorable visual conditions but exhibits unstable and inconsistent concept labeling, often misclassifying tasks despite producing near-correct outputs. The paper identifies four dominant failure modes (concept confusion, overgeneralization, boundary misalignment, and weak object segmentation) and discusses the theoretical implications for hybrid neuro-symbolic architectures. The work is timely, methodologically innovative, and provides valuable diagnostic insights into the reasoning limitations of current LLMs. Its primary weaknesses are the limited scale of the evaluation (10 tasks per family) and the lack of comparison to other models or prompt variations.**

**Rating:** 5
**Confidence:** 4

**Review:**

### Brief Summary of Your Review

This paper presents a targeted evaluation of the DeepSeek V3 large language model on the ConceptARC benchmark, which is designed to test abstract reasoning and conceptual abstraction. The authors introduce a novel evaluation methodology that separates two distinct capabilities: procedural execution (can the model correctly transform an input grid to produce the correct output?) and concept identification (can the model correctly label which type of transformation it is performing?). The key finding is a significant dissociation between these two abilities: DeepSeek V3 achieves moderate success on procedural tasks under favorable visual conditions but exhibits unstable and inconsistent concept labeling, often misclassifying tasks despite producing near-correct outputs. The paper identifies four dominant failure modes (concept confusion, overgeneralization, boundary misalignment, and weak object segmentation) and discusses the theoretical implications for hybrid neuro-symbolic architectures. The work is timely, methodologically innovative, and provides valuable diagnostic insights into the reasoning limitations of current LLMs. Its primary weaknesses are the limited scale of the evaluation (10 tasks per family) and the lack of comparison to other models or prompt variations.

### Detailed Review

**Overview:**
The paper addresses a fundamental question in AI research: to what extent do large language models (LLMs) perform genuine abstract reasoning versus surface-level pattern matching? The authors evaluate DeepSeek V3 on ConceptARC, a benchmark derived from the Abstraction and Reasoning Corpus (ARC) that is designed to measure conceptual abstraction through visual grid transformation tasks. The core innovation of the paper is its methodological separation of two evaluation dimensions:

1.  Procedural Execution: Can the model take an input grid and a transformation description (e.g., "Copy," "Rotate," "InsideOutside") and generate the correct output grid?
2.  Concept Identification: Can the model, given an input-output pair, correctly label which transformation family it belongs to?

By testing both capabilities, the authors can diagnose whether the model has formed a stable, generalizable concept of the transformation or is merely exploiting visual heuristics to produce plausible outputs. The results show a striking gap: procedural accuracy is moderate and highly dependent on visual salience, while concept identification is highly unstable, with frequent misclassifications even when the output is correct. The paper concludes that DeepSeek V3's behavior is best explained by reliance on probabilistic pattern alignment rather than discrete symbolic abstraction, motivating the need for hybrid neuro-symbolic approaches.

**Strengths:**

1.  The separation of procedural execution from concept identification is a significant and insightful contribution. Standard benchmark evaluations typically only measure output accuracy, which can mask fundamental reasoning failures. This dual-task approach provides a much richer diagnostic picture and could become a template for evaluating other models on similar reasoning tasks.
2.  The paper's main result -- a dissociation between "knowing how" and "knowing what" -- is clearly presented and convincingly supported by the per-family analysis in Section 3. The finding that a model can produce a correct output while being unable to correctly name the operation it performed is a powerful demonstration of the limits of current approaches.
3.  Section 4 provides a concise but insightful categorization of failure modes (concept confusion, overgeneralization, boundary misalignment, weak object segmentation). This taxonomy is useful not only for understanding DeepSeek V3's limitations but also for guiding future research and model development.

**Weaknesses:**

1.  The paper evaluates only 10 tasks per family. For a benchmark like ConceptARC, which is designed to probe the boundaries of abstraction, 10 examples per category may not be sufficient to draw robust conclusions about a model's capabilities. A larger sample size would increase confidence in the per-family characterizations (e.g., "Scale is one of the most challenging families").
2.  The study evaluates only one model (DeepSeek V3) with a single prompting protocol. This limits the generalizability of the findings. It is unclear whether the observed dissociation is unique to DeepSeek V3 or a more general property of LLMs. Similarly, the results could be sensitive to prompt wording; a different prompt might yield different concept identification accuracy. The authors acknowledge this as a limitation but do not explore it.
3.  While the per-family qualitative descriptions are useful, the paper lacks a summary table or figure that quantifies the gap between procedural and conceptual accuracy across all families. For example, a scatter plot with procedural accuracy on the x-axis and concept identification accuracy on the y-axis would visually reinforce the core claim of dissociation. The paper would benefit from more aggregated quantitative data.
4.  The paper does not compare DeepSeek V3's performance to any baselines. How does it compare to other LLMs (e.g., GPT-4, Claude)? How does it compare to simpler heuristic or symbolic baselines? Without such comparisons, it is difficult to contextualize the results. Is a 60-70% procedural accuracy on some tasks impressive or underwhelming?
5.  The reference section is extremely sparse. Only two references are provided (Chollet, 2019; Mitchell, 2023). Given that the paper discusses "observations from the ARC literature" and mentions that "neural and prompted systems often succeed at interpolative pattern completion but struggle with systematic compositional generalisation," there should be citations to support these claims. The paper would be strengthened by engaging with the broader body of work on ARC and systematic generalization.

**Suggestions for Improvement:**

1.  Increase the number of tasks per family, if computationally feasible, to improve statistical power.
2.  Evaluate at least one other strong LLM (e.g., GPT-4, Claude 3.5) to see if the dissociation is a general phenomenon.
3.  Test at least one alternative prompting strategy (e.g., few-shot examples, chain-of-thought) to assess sensitivity.
4.  Create a table showing, for each family, the procedural accuracy (output correctness) and concept identification accuracy (label correctness).
5.  Include a visualization (e.g., scatter plot or bar chart) to clearly illustrate the gap between the two capabilities.
6.  Report aggregate statistics (mean, standard deviation) across families.
7.  Compare DeepSeek V3's performance to a simple rule-based system that uses primitive image operations (e.g., flood fill, rotation). This would help establish a lower bound.
8.  If possible, compare to published results of other models on ConceptARC (if any exist).
9.  Add citations to key papers on the Abstraction and Reasoning Corpus (ARC), systematic generalization, and evaluations of LLMs on reasoning tasks. Cite relevant work on neuro-symbolic AI to support the discussion in Section 5.
10.  Provide more detail on how procedural accuracy was measured. Was it exact match? Were there partial credit scores? How were outputs compared to ground truth?
11.  For concept identification, was it a forced choice among the 15 families? Were "none of the above" options allowed? This should be specified.

---

### Decision · Program_Chairs · 2026-03-20

**Decision:**

Accept (Poster)

**Comment:**

Dear Author(s),

On behalf of the Program Committee of the International Conference on Mathematics of Artificial Intelligence (MathAI 2026), we are pleased to inform you that your paper has been accepted for a poster presentation at MathAI 2026.

Your paper was evaluated through a rigorous two-stage review process involving both automated screening and expert review by members of the Program Committee. The reviewers recognized the quality and contribution of your work.

Important Note: The reviewers have recommended final revisions to your manuscript before the conference. Please ensure that all reviewer comments are carefully addressed in your camera-ready version. We trust that you will complete these revisions before the conference deadlines.

Presentation details:

    Format: Poster presentation

    Mode: You may present either in person (offline) at the conference venue in Sirius, Russia, or remotely via Zoom. Please indicate your preferred mode when confirming your participation.

    Conference dates: March 30 - April 3, 2026

    Website: https://mathai.club

Next steps:

    Please confirm your participation and presentation mode by replying to this email (mathai.club@yandex.ru) no later than March 15, 2026 18:00 Moscow time.

    If you plan to attend in person, the organizing committee will provide accommodation details separately.

    Please prepare your final camera-ready manuscript according to the formatting guidelines available at https://mathai.club and upload it to OpenReview by March 15, 2026 18:00 Moscow time. Ensure that all reviewer feedback has been incorporated into this final version.

Should you have any questions regarding the program, logistics, or your presentation, please do not hesitate to contact us.

We look forward to your contribution to MathAI 2026.

With kind regards,

MathAI 2026 Program Committee
International Conference on Mathematics of Artificial Intelligence
https://mathai.club
OpenReview: https://openreview.net/group?id=mathai.club/MathAI/2026/Conference
Telegram: https://t.me/MathAI_club
Email: mathai.club@yandex.ru